# Potential and Limits of Cannabinoids in Alzheimer’s Disease Therapy

**DOI:** 10.3390/biology10060542

**Published:** 2021-06-17

**Authors:** Giulia Abate, Daniela Uberti, Simone Tambaro

**Affiliations:** 1Division of Pharmacology, Department of Molecular and Translational Medicine, University of Brescia, 25123 Brescia, Italy; giulia.abate@unibs.it (G.A.); daniela.uberti@unibs.it (D.U.); 2Department of Neurobiology, Care Sciences and Society, Center for Alzheimer Research, Division of Neurogeriatrics, Karolinska Institutet, 171 64 Solna, Sweden

**Keywords:** Alzheimer’s disease, cannabinoids, THC, cannabidiol, CB1, CB2, anandamide, 2-AG, amyloid-β, FAAH

## Abstract

**Simple Summary:**

This review was aimed at exploring the potentiality of drugging the endocannabinoid system as a therapeutic option for Alzheimer’s disease (AD). Recent discoveries have demonstrated how the modulation of cannabinoid receptor 1 (CB1) and receptor 2 (CB2) can exert neuroprotective effects without the recreational and pharmacological properties of *Cannabis sativa*. Thus, this review explores the potential of cannabinoids in AD, also highlighting their limitations in perspective to point out the need for further research on cannabinoids in AD therapy.

**Abstract:**

Alzheimer’s disease (AD) is a detrimental brain disorder characterized by a gradual cognitive decline and neuronal deterioration. To date, the treatments available are effective only in the early stage of the disease. The AD etiology has not been completely revealed, and investigating new pathological mechanisms is essential for developing effective and safe drugs. The recreational and pharmacological properties of marijuana are known for centuries, but only recently the scientific community started to investigate the potential use of cannabinoids in AD therapy—sometimes with contradictory outcomes. Since the endocannabinoid system (ECS) is highly expressed in the hippocampus and cortex, cannabis use/abuse has often been associated with memory and learning dysfunction in vulnerable individuals. However, the latest findings in AD rodent models have shown promising effects of cannabinoids in reducing amyloid plaque deposition and stimulating hippocampal neurogenesis. Beneficial effects on several dementia-related symptoms have also been reported in clinical trials after cannabinoid treatments. Accordingly, future studies should address identifying the correct therapeutic dosage and timing of treatment from the perspective of using cannabinoids in AD therapy. The present paper aims to summarize the potential and limitations of cannabinoids as therapeutics for AD, focusing on recent pre-clinical and clinical evidence.

## 1. Introduction

Alzheimer’s disease (AD) is one of the principal conditions of disability among older people, which impairs a person’s ability to function in daily life. Currently, it is estimated that more than 50 million people are suffering from AD worldwide [1]. Furthermore, since one of the main risk factors of AD is aging, and the human lifespan is constantly increasing, the number of AD cases is projected to double in the following decades [1]. AD can be divided based on its pathophysiology in sporadic or late-onset AD and familial or early-onset AD. Sporadic AD, the preeminent form of AD (about 95% of all cases), is a multifactorial disease, where the etiopathogenesis is still not fully understood and is influenced by epigenetic and genetic variants combined with environmental and lifestyle factors. In contrast, familial AD is rare (<5%) and is caused by gene mutations of amyloid precursor protein (APP) and presenilin-1 and 2 (PSEN1 and PSEN2) [2]. Both sporadic and familial AD develop a similar pathology consisting of parenchymal deposition of amyloid-β (Aβ) in plaques and intraneuronal accumulation of hyperphosphorylated tau protein, leading to brain inflammation and oxidative stress that have a fundamental impact on the onset of the disease [3,4,5]. To date, there is no effective cure, and the treatments available can reduce only the symptoms in the initial phase of the disease. For that reason, it is of paramount importance to identify novel effective compounds for counteracting the AD course or even treat the disease [6,7]. Therefore, a better understanding of the etiopathological mechanisms involved in AD may provide novel effective, druggable targets for AD treatment.

The Endocannabinoidergic System (ECS) plays an essential role in brain memory and cognitive function in multiple ways, and most importantly, ECS is involved in synaptic responsiveness and plasticity [8]. The high presence of the primary ECS receptor, the cannabinoid receptor 1 (CB1) in the hippocampus and cortex, seems to be the main factor responsible for the psychotropic and cognitive effects linked to cannabis use. Controversial side effects have been observed after marijuana and synthetic cannabinoids exposition [9,10]. Learning and memory impairment has been reported in several studies, especially in young individuals [11,12]. Since brain development is completed only around the age of 25, cannabis use in adolescence may be associated with increased adverse effects on brain formation and function, particularly in areas sensitive to the pharmacological effects of cannabis. However, over recent decades the modulation of the ECS has emerged as a potentially attractive strategy for treating AD. Activation of both CB1 and cannabinoid receptor 2 (CB2) has revealed beneficial neuroprotective effects reducing β-amyloid deposition and tau phosphorylation. It should also be noted that low doses of Δ^9^-tetrahydrocannabinol (THC) showed several beneficial effects by inducing hippocampal neurogenesis and reducing Aβ toxicity (i.e., plaque deposition) in rodents, as well as in other dementia-related symptoms in both pre-clinical and clinical studies [13,14]. Furthermore, (i) the phytocannabinoid cannabidiol, (ii) the activation of the CB2 receptors, and (iii) the modulation of the endogenous cannabinoid levels all seem to be potentially attractive strategies for the absence of psychoactive effects, instead observed, after stimulation of the CB1 receptor [15,16,17]. Several synthetic selective CB1/CB2 agonists/antagonists and inhibitors of endogenous cannabinoid degradation have been generated and tested for their therapeutic effects in the last years. Thus, this review aims to summarize the most recent advances in cannabinoids research for AD, describing their limitations and potential as a therapeutic option.

## 2. Cannabinoids and Endocannabinoid Systems

The recreational and pharmacological properties of marijuana have been known since ancient times. The first texts documenting the medical benefits of marijuana dates back to a Chinese medical manual from approximately 2700 B.C. [18]. In recent decades, the scientific community has deepened its investigation of the chemical properties of the principal actives in marijuana extract, yet recently, the attention has been focused on understanding the biological mechanism involved in their multifaceted effects [19]. Marijuana (or *Cannabis sativa*) contains more than 500 distinct compounds, where 120 are classified as phytocannabinoids with different chemical structures and pharmacological properties [20]. The first compounds isolated from marijuana extract were cannabinol (CBN) and cannabidiol (CBD) in 1940 [21], followed years later in 1964 by the isolation of the main psychoactive component of marijuana (−)-*trans*-Δ^9^-tetrahydrocannabinol (Δ^9^-THC or THC) [22]. A milestone in the modern history of the therapeutic use of cannabis is associated with the identification of the endocannabinoid system in the early 1990s [23]. The isolation, cloning, and expression of the CB1 receptor were succeeded some years later with the characterization of the CB2 receptor [24]. Both these receptors are coupled to the Gi/o proteins signal transduction pathway. Over recent years, several other receptors have been associated as part of the endocannabinoidergic system and were able to modulate the effect of phyto- and synthetic cannabinoids and endogenous ligands, such as the orphan G protein-coupled receptors, GPR3, GPR6, GPR12, and GPR55, and the nuclear hormone peroxisome proliferator-activated receptors (PPARs) [25,26,27,28]. The CB1 receptor is expressed in both the peripheral and central nervous systems, where it is predominantly presynaptically located (Figure 1). The brain distribution of CB1 is consistent with the known physiological effects of cannabinoids as impairment of short-term memory formation, altered motor activity, and anxiety [29]. High levels of CB1 receptors have been detected in the hippocampus, cortical regions, and the cerebellum. Only recently studies have reported the presence of CB1 receptors in astrocytes [30,31,32], where CB1 activation was associated with an increase in calcium uptake and release of glutamate. On the contrary, the CB2 receptor is, for the most part, expressed in the peripheral immune system cells and tissues. The presence in the brain of CB2 is very low compared to CB1 and has been detected in the ventral tegmental area and hippocampal neurons [33]. Nevertheless, CB2 seems to play a crucial role in macrophage/microglia functions [34,35]. The expression of CB2 drastically increases in activated microglia, and activation of CB2 decreases the production of proinflammatory molecules [36]. Another important event in revealing the brain cannabinoidergic system was the isolation of endogenous compounds, which were able to modulate the cannabinoid receptors. The most investigated and characterized are the arachidonic acid derivatives: *N*-arachidonoylethanolamine (anandamide or AEA) and 2-arachidonoylglycerol (2-AG) [37,38]. A particularity of endocannabinoids is that they are produced postsynaptically on demand and are not stored in vesicles [39]. As described in Figure 1 that reported a schematic representation of the endocannabinoidergic system at the neuronal level, endocannabinoids are released in the synaptic cleft from the postsynaptic neurons. They interact with the cannabinoid receptors located on the presynaptic neurons, negatively modulating the GABA and glutamate release [40]. Anandamide and 2-AG have a very short half-life. After their secretion in the synaptic cleft, these compounds re-uptake and are hydrolytically inactivated by the integral membrane enzyme fatty acid amide hydrolase (FAAH) and the monoacylglycerol lipase, respectively (MAGL) [41,42]. Most remarkably, the release of anandamide and 2-AG in the brain affects memory, memory acquisition, and consolidation such as long-term potentiation [43].

### 2.1. Phytocannabinoids and Modulation of Cannabinoid Receptor 1 (CB1)

The CB1 receptor is one of the most abundant G protein-coupled receptors present in the brain. In humans, it is mainly expressed in the hippocampus, cortex, basal ganglia, brainstem, and cerebellum [44]. The high presence of CB1 in the hippocampus and cortex correlates with the documented effect of cannabinoids on the learning and memory process. Although the pathophysiological role of CB1 in AD is still elusive, the lack of CB1 receptors has been associated with a faster decline of cognitive function and loss of neurons in the hippocampus in WT mice [45]. Lee and colleagues (2010) [46] demonstrated that CB1 receptor levels do not change in AD, and they suggested a role of CB1 in preserving cognitive function. Interestingly, CB1, together with the CB2 cannabinoid receptor, was found in Aβ plaques in post-mortem brain tissue from individuals with AD [47]. Several findings showed that acute activation of CB1, especially at a young age, negatively affects dose-dependently short-term memory performance [48,49]. An analogous consequence has also been reported for chronic users, through observation, a decrement in the capacity to learn and remember new information compared to non-marijuana users [50]. In contrast, there is no clear evidence that acute or chronic use of cannabis has a permanent impairment in long-term memory and working memory [51]. Even though undesired psychoactive effects have conditioned the medical research and have created skepticism in the therapeutic use of cannabis and its related chemicals, a consistent beneficial impact in memory impairment in AD-aged rodents and humans has been described for THC, cannabidiol, and other synthetic compounds. Findings that endorse the CB1 receptor as a potential therapeutic target for AD treatment and it needs and deserves further investigation.

### 2.2. THC

Δ^9^-THC or THC is the most abundant compound among the more than 500 components isolated from marijuana extract [52] and is the primary psychoactive component of cannabis. THC has a similar affinity for both CB1 and CB2 receptors, although most of the THC psychoactive effects are related to the activation of CB1 receptors [53].

Chronic and acute intoxication by marijuana, and consequently to THC, has often been associated with several adverse effects, such as a reduction in most cognitive functions, learning, memory, attention, and executive function [54], and in some vulnerable subjects, an increased risk of both psychotic symptoms and schizophrenia-like psychoses [55]. In cannabis-dependent subjects, a deficit in striatal dopamine release was found [56], and a single-photon emission computed tomography analysis showed lower hippocampal perfusion among marijuana users than controls [57]. Furthermore, abnormalities in axonal connectivity and hippocampus and amygdala volumes have been found in long-term, heavy cannabis users [58,59]. The recreational consumption of cannabis has increased in the past few years, and particularly in those countries that have legalized the use as well as reduced the starting age for consumers. With the increased potency of cannabis in the last few years, vulnerable users being negatively impacted has ensued. There has been a continual increase in the THC content or potency of marijuana in recent decades, from approximately 4% in 1995 to 17% in 2017 [60]. This rise increases the chance of experiencing adverse effects linked to recreational consumption [61].

Conversely, the use of marijuana and THC have shown a strong therapeutic potential for the treatment of neuronal inflammation and neurodegenerative diseases such as AD. For the correct interpretation of the therapeutic potential of THC, it is vital to circumscribe and separate the THC or marijuana effects reported under a “non-medical” (recreational consumption), and list it under recreational consumption along with the pre-clinical findings and the effects reported in clinical trials under medical supervision. Essentially, the primary reported harmful effects of marijuana came from studies conducted in young adults or in adolescents, which is a critical period of development associated with a high vulnerability to the central effects of the drugs, whereas few studies have been conducted in adults. Only recently, a biphasic dose-response and an age-related effect started to be considered important discriminative factors to induce a beneficial impact of THC on the brain and cognition [62] (Figure 2). THC showed a broad spectrum of effects that could be potentially beneficial in blocking or preventing AD. For example, THC has shown an anti- Aβ aggregation activity in an in vitro study. THC reduced the fluorescence intensity in the thioflavin test in a dose-dependent manner by direct interaction with the Aβ peptide [63], affecting Aβ fibril formation and aggregation [64]. THC stimulates the removal of intracellular Aβ and blocks the inflammatory response [65,66]. THC has been shown to inhibit the enzyme acetylcholinesterase (AChE) activity more effectively than the approved drugs for AD treatment—donepezil and tacrine [67]. In rat cortical neuron cultures, the toxicity induced by high levels of the excitatory neurotransmitter glutamate was inhibited by THC. The neuroprotection effects of THC were not reduced by cannabinoid receptor antagonist, indicating a therapeutic mechanism not mediated by cannabinoid receptors [68]. Administration of low doses of THC in rats was associated with enhanced neurogenesis in the brain, especially in the hippocampus, and an improvement of cognitive functions. The administration of ultralow doses of THC in mice protected the brain from LPS neuroinflammation-induced cognitive damage [69]. THC was effective in significantly reducing Aβ levels and neurodegeneration in 5XFAD transgenic mice by increasing the levels of neprilysin, the endopeptidase responsible for Aβ degradation [70]. In APP/PS1 mice treated with THC, astrogliosis, microgliosis, and inflammatory-related molecules were found reduced with effects that were even stronger in the combined treatment of THC and CBD [71]. Chronic treatment with THC and CBD improved memory impairment in APP/PS1 mice at advanced stages of the AD pathology. However, this treatment did not change the Aβ deposition and gliosis; phenomena instead observed when THC and CBD were administered at the early stages of the disease [72]. The therapeutic effects produced by THC and CBD in aged APP/PS1 mice were combined with an improvement of synaptic function. In particular, the treatment induced a reduction in metabotropic glutamate receptor 2/3 and increased the levels of GABA-A Rα1 compared with control mice [72]. In general, CB1 receptor agonists and THC induced the release of brain-derived neurotrophic factor BDNF in cells and several brain regions [73]. This phenomenon can be one of the main biological events linked to the THC neuroprotective effect. In this respect, Marsicano et al. [74] revealed that the CB1-induced BDNF expression participates in the therapeutic effect of CB1 receptor activity against neurotoxicity. These results have a high translational value considering that BDNF signaling regulates morphological and physiological synaptic plasticity. Most importantly, BDNF expression declines with aging and even more in pathological aging, and re-established BDNF physiological levels can be considered an essential way for rescuing synaptic plasticity in AD patients.

On the clinical side, in patients with AD treated for six weeks with dronabinol (2.5 mg), a synthetic form of THC was observed as a positive effect on body weight and an improvement in disturbing behavior [14]. After two weeks of treatment, dronabinol (2.5 mg) reduced nighttime activity and agitation in patients in advanced stages of AD [75]. In another study, low-dose oral THC (1.5 mg)—in a 21-day-treatment—did not affect dementia-related neuropsychiatric symptoms. In contrast, it was tolerated in treated patients, and no relevant side effects were reported [76]. THC safety at low concentration and rapid absorption (with maximum plasma concentrations at two hours after treatment) was also reported in another clinical study on older dementia patients [77]. Efficacy and safety with a significant reduction in the Neuropsychiatric Inventory and Clinical Global Impression severity scale was also reported in patients treated with medical cannabis oil (MCO) containing THC [78]. The synthetic oral THC analog Nabilone significantly reduces agitation over six weeks of treatment in AD patients. Nabilone was also associated with significant improvements in overall neuropsychiatric symptoms and caregiver burden [79].

The clinical efficacy of THC on agitation and aggression in patients with AD remains inconclusive, though there may be a signal for a potential benefit of synthetic cannabinoids.

### 2.3. Cannabidiol

The other main phytocannabinoid in cannabis plants is cannabidiol (CBD), which comprises up to 40% of the total compounds extract. CBD, as opposed to THC, has no psychotropic properties, as also confirmed in a recent trial where healthy volunteers did not show any effects in the emotional state, cognitive performance, or attention after receiving CBD [80]. CBD has a very low affinity to the CB1 and CB2 receptors [53], and several findings proposed that CBD operated as a negative allosteric modulator/inverse agonist in both CB1 and CB2 receptors [81,82,83]. Furthermore, CBD acts as an inverse agonist for G protein-coupled orphan receptors such as GPR3, GPR6, and GPR12. Other studies reported that CBD could activate the Transient Receptor Potential Vanilloid (TRPV) channels, serotonin (5-HT1A), PPARs, *N*-methyl-D-aspartate (NMDA) receptor, and α-amino-3-hydroxy-5-methyl-4-isoxazolepropionic acid (AMPA) receptors. The potential employment of CBD in AD therapy is under debate, with still few studies available. However, several findings support the therapeutic potential of this compound in improving some symptoms associated with AD. Notably, in preclinical characterization, CBD exhibited neuroprotective, anti-inflammatory, anxiolytic, and anti-insomnia properties. In this context, the strong antioxidant effect of CBD was reported against glutamate toxicity in primary neuronal culture [68]. The neuroprotection and antioxidant properties of CBD were also observed on β-amyloid peptide-induced toxicity in cultured rat PC12 cells [84]. CBD modulates microglial cell function in vitro and prevents the learning of a spatial navigation task and TNF-α and IL-6 gene expression in β-amyloid-injected mice [71,85]. Tau hyperphosphorylation plays a crucial role in the pathogenesis of AD. In this regard, it has been demonstrated that CBD inhibits β-amyloid-induced tau protein hyperphosphorylation nitric oxide production [86,87]. In mesenchymal stem cells treated with CBD, a lower gene expression of some specific genes associated with AD were observed, including genes coding for the proteins responsible for tau phosphorylation and Aβ production as the β- and γ-secretase genes [88]. Likewise, CBD prevented the expression of proinflammatory glial molecules in the hippocampus of an in vivo model of Aβ-induced neuroinflammation. CBD prevented the expression of proinflammatory glial peptides in the hippocampus of mice Aβ-induced neuroinflammation [89]. Long-term oral CBD treatment improved the social recognition memory and pathophysiology of a double transgenic APP × PS1 mouse model for AD [90]; in the same mouse model, CBD treatment significantly up-regulated the autophagy pathway [91]. Furthermore, in a recent case study, CBD consumption significantly improved neuropsychiatric symptoms in AD patients [92].

### 2.4. Synthetic CB1 Modulators

Several synthetic cannabinoid compounds have been generated in the last decades with the aim to selectively investigate the physiological and pathophysiological role of the two primary endocannabinoid receptors CB1 and CB2. These synthetic compounds have been tested as a therapeutic tool in several pre-clinical models, including in vivo and in vitro AD models. To date, the only chemical modification of Δ^9^-THC that has reached the status to be an approved drug from the FDA is nabilone, under the name Cesamet, for the treatment of nausea and vomiting associated with cancer chemotherapy [93]. CP 55,940 was the first synthetic cannabinoid analog to be synthesized from a chronological perspective [94], followed by several others. Some of the most extensively studied selective CB1 or mixed CB1/CB2 agonists are WIN 55,212-2, HU 210, ACEA, and JWH-018. Regarding these compounds, in recent decades, numerous preclinical studies in rodents, despite sometimes controversial, have highlighted their positive effects on memory and learning processes and on other neurobiological mechanisms underlying AD. Systemic administration of CP 55,940, WIN 55,212-2, and ACEA affected working memory [95] and object recognition memory in rats [96,97]. A similar effect using CP 55,940 was also reported in mice [98]. The negative effects of synthetic cannabinoids (WIN 55,212-2 and CP 55,940) on learning and memory appear to be directly linked to the inhibition of acetylcholine release in the hippocampal region [99,100] and the inhibition of glutamatergic synaptic transmission in the prefrontal cortex [101,102]. Nonetheless, CB1 receptor modulation in the hippocampus is essential for the memory disruptive effects of cannabinoids but are not essential for the other common CNS actions [103]. Hippocampal slices exposure to synthetic cannabinoid agonists (WIN 55,212-2, HU 210) affects long-term potentiation (LTP) [104,105], and (HU 210; JWH-018) alter spontaneous firing, bursting, and synchronicity in hippocampal cells [106,107,108]. Acute administration in mice of JWH-018, known in the illegal market as *Spice* and ‘herbal blend’, impair cognitive function affecting hippocampal synaptic transmission and memory mechanisms [108]. A decrease in BDNF following JWH-018 treatment was observed in the hippocampus. As previously mentioned, the neurotrophic factor BDNF plays an important role in modulating the learning and memory process, promoting neurogenesis, synaptogenesis [109], and the alteration of BDNF levels after JWH-018 exposition, which may explain its negative effect on memory performance. Nevertheless, this effect on BDNF release observed with JWH-018 is in contradiction with the effect reported previously with THC administration, where an increased BDNF production was observed. Together with the negative effect on memory and learning processes, other findings strongly supported a beneficial therapeutic effect by CB1 receptors activation. Likewise, reported for the phytocannabinoid THC, neurogenesis in the hippocampus of aged rats could be induced using a low dose of WIN 55,212-2 [110]. A similar effect was also observed after chronic treatment with HU 210, which promoted neurogenesis in the dentate gyrus of adult rats [111]. The primary role of CB1 cannabinoid receptors in regulating neurogenesis in the adult brain was confirmed in CB1-knockout mice, which showed reductions in the number of BrdU-labeled cells to −50% of WT levels in the dentate gyrus and subventricular zone—suggesting that CB1 activation promotes neurogenesis. The involvement of CB1 in neurogenesis was further confirmed in CB1 knockout mice where it was observed defective adult neurogenesis [112]. Furthermore, the treatment of activated primary human astrocytes with WIN 55,212-2 significantly reduced in a dose-dependent manner the expression and release of cytokines [113]. HU 210 ameliorated the memory deficits of olfactory bulbectomized (OBX) rats [114]. In contrast with what was previously reported, chronic treatment with WIN 55,212-2 significantly normalizes this cognitive deficit in old Tg APP mice accompanied by a reduction in the inflammation and an increased Aβ clearance [115,116]. Chronic administration of the selective CB1 agonist ACEA at pre-symptomatic or early AD stages reduced the learning and memory deficits observed in the double APP/PS1 transgenic mice. In primary neuronal cell cultures, ACEA reduced the cytotoxic effect induced by Aβ42 oligomers and reduced Aβ-induced glycogen synthase kinase-3β activity in cortical neurons. Moreover, a defect in astroglial response and a decreased expression of the proinflammatory interferon-gamma were found in the surrounding area of Aβ plaques deposition in ACEA-treated mice when compared with non-AD mice [117]. The infusion of ACEA in the rat hippocampus prevented the neurotoxic Aβ-induced effect. ACEA prevented cognitive impairment and decreased the activation of microglia and astroglia in the dentate gyrus [118].

### 2.5. Modulation of Cannabinoid Receptor 2 (CB2)

The other side of the endocannabinoid system is mainly represented by the CB2 receptor—mostly considered as related to the periphery—as it was initially found to be highly expressed at the spleen level and hard to detect in the brain. Today, many confirmed CB2 expression in selective areas of the brain, despite its main localization in the microglia. In detail, Svizenska et al. [119], mapping the CB2 receptor distribution in the mammalian nervous system, found CB2 receptor in the anterior olfactory nucleus in the neurons of the piriform, orbital, visual, motor, and auditory cortex. However, CB2 receptors in physiological conditions are expressed very low in the brain while increasing in the expression in both neuronal and non-neuronal cells but only in pathological conditions. CB2 receptors may play a role in nociception [120,121], gastrointestinal function [122], neural progenitor cell proliferation and axon guidance [123,124], and synaptic transmission [125,126] among other functions.

Since CB1 receptors are primarily related to the unwanted psychotropic effects of marijuana-derived cannabinoids, the CB2 receptor becomes really attractive as a druggable target. The potential therapeutic use of CB2-agonist in AD is also reinforced by the findings that in the AD human brain, CNR2 (the gene encoding the CB2 receptor) was found to be increased compared to age-matched controls [127]. The anti-inflammatory effects of CB2 agonists have been widely described in different transgenic mouse models of AD and in in vitro AD-like models [128]. Additionally, it was demonstrated that in Aβ-treated mice, cannabinoid treatment prevented microglial activation and avoided induced cognitive impairment. In human postmortem AD brain tissues, cannabinoid CB2 receptors were found selectively overexpressed in neuritic plaque-associated glia [129]. A novel CB2 agonist (MDA7) promised improved cognitive performance in rats microinjected with Aβ into the hippocampus by favoring Aβ clearance [130]. CB2 receptors, as reported for CB1, are involved in neurogenesis. In fact, in CB2-deficient mice, the number of BrdU+ cells in the dentate gyrus was found reduced [15,123].

Evidence suggests that neuroinflammation may be pivotal in tangle formation [131]. Thus, another therapeutically CB2-mediated effect was also linked to the modulation of hyperphosphorylated tau, another benchmark of AD. In fact, chronic administration of JWH-133, a selective CB2 receptor agonist, was found effective in reducing tau hyperphosphorylation surrounding Aβ plaques in APP/PS1 mice [132]. Furthermore, mice overexpressing human tau (PK−/−/TauVLW) showed a marked reduction in neurofibrillary tangles with prolonged treatment with Sativex^®^, an already approved medicine based on mixed Δ9-THC and CBD natural extracts [133]. Due to the multifactorial and sporadic nature of AD, multi-target drugs capable of acting on multiple targets simultaneously (comprising the CB2 receptor) are becoming an attractive therapeutic option in the field of AD. Recently, Scheiner et al. [134] synthesized dual-acting hybrid compounds combining the effects of a benzimidazole-based CB2 selective agonist with those of tacrine as a cholinesterase (ChE) inhibitor. These hybrids showed neuroprotection against glutamate-induced oxidative stress when tested in vitro while showing pronounced effects on short- and long-term memory, avoiding the hepatotoxicity side effect of tacrine [134]. Again, with a similar hybrid approach, Montanari et al. [135] identified a potent and selective hybrid CB2-ligand able to simultaneously restore the cholinergic system by inhibiting butyrylcholinesterase (BuChE), within addition neuroprotective activity against Aβ1-42 oligomers and immuno-modulatory effects, addressing microglia to the neuroprotective M2 phenotype [135]. Consequently, multi-target CB2 agonists can be useful in the development of neuroprotective and potential immunomodulating drugs for AD, acting via the endocannabinoid system.

### 2.6. Modulation of Endogenous Cannabinoid Anandamide and 2-AG

Modulating the levels of the endogenous cannabinoid compounds (i.e., anandamide and 2-AG by pharmacological blockade of their degradation) is a potential therapeutic approach for treating AD. The inhibition of the two main endocannabinoid hydrolase enzymes, FAAH and MAGL, augments the levels of endocannabinoid available for interaction with their receptors. Most importantly, it augments no relevant undesirable side effects in motility, catalepsy, body temperature, or cognition as reported for high doses of CB1 agonists [136,137,138]. Specifically, relevant expression changes of anandamide (2-AG) and their proteolytical enzymes (FAAH and MAGL) during normal aging and the neurodegenerative process have been observed in both humans and rodents [139,140]. 

In humans, the analysis of frontal and temporal cortex tissues from post-mortem AD patients revealed significantly reduced levels of anandamide compared to the control subjects. Yet, no differences in 2-AG levels were observed [141]. Moreover, anandamide levels have been inversely correlated with Aβ42 but not with Aβ40, amyloid plaque deposit, or tau protein phosphorylation. In another study, on the contrary, in the frontal cortex from human AD patients and in aged-rat synaptic terminals, a higher anandamide availability and reduced FAAH synaptic activity were observed [142]. High levels of the anandamide hydrolase enzyme FAAH were instead found around the amyloid plaque deposition in astrocyte and microglia cells, supporting the ECS may play a modulatory role in the inflammatory response in the AD neuroinflammation process surrounding the plaques [129]. The hippocampal protein concentrations for the DAGLα and DAGLβ, 2-AG-biosynthesizing enzymes, were also found to be significantly increased in the advanced stage of AD (Braak stage VI) in microglia accumulating near senile plaques [143]. 

In rodents, an enhancement in the mRNA levels of the 2-AG-biosynthesizing enzyme DAGLα together with a higher level of 2-AG was also observed at hippocampal level after acute stereotaxic injection of amyloid proteins into the rat cortex [144]. In the same study, the β-amyloid-induced neuronal toxicity in the hippocampus was reversed by VDM-11, an inhibitor of endocannabinoid cellular reuptake. In WT mice, hippocampus 2-AG, but not anandamide levels, decreased during aging; this decrease seemed to be linked with a significant reduction in DAGLα expression at both protein and mRNA levels and by enhanced MAGL activity [140]. Still, there is a lack of information concerning the age-related changes in endocannabinoid levels, and more research is needed to clarify some controversial findings reported in the literature. However, re-establishing the physiological endocannabinoid tone may represent a preventive or even a potential treatment for AD. 

In this context, the selective pharmacological inhibition of FAAH and MAGL or dual inhibition of FAAH/MAGL—with the following increase in anandamide and 2-AG—promotes a reduction in Aβ-protein deposition in an AD rodent’s model. Currently, several classes of reversible and irreversible covalent FAAH inhibitors have been developed, such as URB597, OL-135, PF-3845, AM3506, PF-04457845, JNJ-40355003, JNJ-42165279, JNJ-1661010, and BIA 10-2474, although the majority of studies have involved URB597. The irreversible covalent URB597 promoted the increase in endocannabinoid anandamide by inhibiting FAAH activity [145,146]. Furthermore, URB597 efficiently suppressed glutamate Aβ42-induced toxicity in primary hippocampal neurons and stimulated the mitochondrial membrane potential [147]. URB597 treatment is associated with the reduction in interleukin (IL)-1β, tumor necrosis factor-α (TNFα) expression, and restoration of long-term potentiation in aged rats [148]. Similar findings have been recently reported after treating monocytes/macrophages from AD patients with URB597, where a general reduction in proinflammatory cytokines was observed [149]. It is noteworthy to mention the inhibition of FAAH by OL-135 accelerated acquisition and extinction rates in a spatial memory task [150]. Many FAAH inhibitors (i.e., PF-04457845 and JNJ-42165279) have been characterized mainly for their analgesic and anxiolytic effects in rodents and humans [138,151]. In particular, PF-04457845, which is 25-fold higher for human FAAH inhibition than URB597, showed a high in vivo efficacy and long duration of action in a rat model of inflammatory pain with also a high oral bioavailability and high brain penetration. These results made this compound a strong candidate to be used in the clinical treatment of central nervous system disorders. So far, only a few clinical trials exist; however, the pharmacological effects of PF-04457845 have been evaluated in humans and were found to be very well tolerated in healthy subjects [152]. It should also be mentioned that the inhibition and the knockdown of FAAH suppressed prostaglandin E2 production and proinflammatory gene expression [153] supported even stronger FAAH inhibition as a therapeutic strategy for reducing AD-related neuroinflammation. The effects of selective inhibition of MAGL have also been characterized. The MAGL inhibitors synthesized can be classified into irreversible inhibitors (maleimides, disulfides, carbamates, ureas, and arylthicarmide) and reversible inhibitors (tetrahydrolipstatin-based derivatives, isothiazolines, natural terpenoids, and amide-based derivatives). Pharmacological and genetic inactivation of MAGL (in a mouse model of AD) attenuated eicosanoid levels, attenuated glial activation and associated neuroinflammation, lowered amyloid β levels, and reduced amyloid plaque burden [154]. Interestingly, a reduced prostaglandin production, rather than enhanced endocannabinoid signaling, seemed to be the underlying main pathophysiology mechanism involved. In this regard, MAGL has been shown that, with the hydrolyzes of 2-AG, it generates the primary arachidonic acid pool for neuroinflammatory prostaglandins [155]. Among different MAGL inhibitors, JZL184 was characterized first, and then after further structural modification, several new derivatives of JZL184 were generated [17]. In an AD mouse model where JZL184 was used as a treatment, a decrease in proinflammatory reactions of microglia, along with reduced total Aβ burden and its precursors, were found. Likewise, it reduced the proinflammatory responses of microglia and astrocytes isolated from adult mice [156]. Inhibition of MAGL enzyme activity and subsequent increase in 2-AG correlated with decreased Aβ accumulation and expression of β-secretase (or BACE1), an enzyme involved in APP cleavage and Aβ generation. MAGL inhibition has been associated with several anti-AD effects: reducing neuroinflammation, improving synaptic plasticity, spatial learning, and memory in AD animals [8]. 

The compound JZL195 is a potent inhibitor of both FAAH and MAGL, with an IC50 of 2 and 4 nM, respectively [17]. Subcutaneous delivery of JZL195 enhanced the brain levels of anandamide and 2-AG in a concentration-dependent way and produced anti-allodynic effects in a mice model of chronic neuropathic pain [17,157]. The important role of the endocannabinoid system in the adult neurogenesis process was confirmed in FAAH-deficient mice [158]. In these mice, the hippocampal proliferation of multipotent neural progenitor cell counting was significantly higher when compared with control WT mice. A similar finding was also observed increasing the levels of anandamide by pharmacological inhibition of FAAH activity [159]. Additionally, in DAGL-KO mice, the adult neurogenesis in the hippocampus and the subventricular zone was compromised [160]. Although the molecular mechanisms responsible for the FAAH and MAGL effects against neuropathology of AD remain to be determined, the findings reported so far support that FAAH and MAGL would be promising therapeutic targets for preventing and treating AD. Therefore, the pharmacological inhibition of these two enzymes has appeared as a potentially appealing strategy to elevate endocannabinoidergic tone. Table 1 summarizes the principal AD-related beneficial and adverse effects of the prevalent cannabinoids described. 

Another potential of endocannabinoids as a therapeutic option for AD is their ability to modulate the mammalian target of the rapamycin (mTOR) signaling pathway [161,162]. The activation of mTOR is a trigger for Aβ generation; thus, its inhibition is an important therapeutic target for AD [163]. Of note, 2-AG treatment was able to prevent the activation of mTOR signaling pathway in the hippocampus in mice through a CB2-dependent mechanism [164]. Again, CB1 and mTOR are intimately linked and involved in regulating excitatory glutamatergic inputs and energy balance at the brain level [165]. Overall, despite this intriguing link between endocannabinoids and mTOR need to be further explored, these data further confirmed the endocannabinoid system as an attractive therapeutic strategy to be further deepened in AD. 

**Table 1 biology-10-00542-t001:** Cannabinoids principal AD-related beneficial and adverse effects.

Compounds	Endocannabinoid System Targets	Beneficial Anti-AD Effects	Adverse/Unwanted Effects
THC	Mixed CB1 and CB2 agonist	Inhibition of achetylcholinesterase [67] Reduce Aβ levels [63] Hippocampal neurogenesis [166] Induce BDNF release [73,74]	Psychotic effects [55] Reduce cognitive functions [54] A deficit in dopamine release [56]
CBD	Mixed CB1 and CB2 agonist	No psycoactive effets [80] Neuroprotection [84] Reduce microglia activation [85] Delay cognitive decline [167]	Hypotension at high doses [168] Anxiogenic-like effect [169]
WIN 55,212-2 HU 210 CP 55,940 JWH-018	Mixed CB1 and CB2 agonist	Increase Aβ clearance [116] Promote neurogenesis [111] Prevent cognitive impairment [113,114]	Defect in working memory [95,96,97] Affects long-term potentiation [104,105] Sedation [170]
ACEA	Selective CB1 agonist	Anti-inflammatory [117] Prevent spatial memory impairment [118]	N.R.
JWH-133 AM-1241 MDA7	Selective CB2 agonist	Increase Aβ clearance [116] Improve cognitive performance [116] Prevent microglial activation [128] Reduce tau hyper-phosphorylation [132]	Immune suppression [171]
URB597 PF-04457845 JZL184 JZL195	Modulation of endogenous cannabinoid anandamide and 2-AG	Suppress glutamate Aβ42-induced toxicity [147] Reduce proinflammatory interleukin expression [148,156] Restore long-term potentiation [148] Reduce amyloid plaque burden [154]	Cardiac diastolic stiffness [172]

## 3. The Orphan G Protein-Coupled Receptors (GPRs)

In addition to the two well-characterized G protein-coupled receptors CB1 and CB2, several orphan G protein-coupled receptors or GPRs have been described in the last years to be putative cannabinoid receptors, such as GPR3, GPR6, GPR12, GPR18, and GPR55 [173,174,175]. GPR3, GPR6, and GPR12 have a close phylogenetic affinity and conserve specific sequences with the cannabinoid receptors CB1 and CB2 [173]. Moreover, these receptors are highly expressed in several brain areas, where sphingosine 1-phosphate (S1P) and sphingosylphosphorylcholine have been identified as putative endogenous ligands of GPR3, GPR6, and GPR12 [176,177]. Recently, it has also been shown that CBD, the non-psychotropic phytocannabinoid, binds to GPR3, GPR6, and GPR12, acting as an inverse agonist [27]. Therefore, besides their physiological role is still unclear, they seem involved in several brain processes related to pain, memory, and emotion. Interestingly, GPR3 was found highly expressed in the AD postmortem brain and correlated with the entity of the AD pathology [178]. The activation of GPR3 directly affects Aβ-plaques deposition by stimulating Aβ production [179]. On the other hand, genetic deletion of GPR3 decreased the amyloid plaque deposition and improved cognitive impairment in preclinical AD mouse models [178]. An increased hippocampal expression was also observed for GPR6 in the 3 × Tg AD mouse model, where GPR6 modulates the neuroprotective effect of the complement protein C1q against Aβ [180]. GPR18 and GPR55 despite, GPR3, GPR6, and GPR12, have low homology with CB1 and CB2 [175]. GPR55 has opposite signaling pathways from CB1/CB2 since it is coupled to Gα_12,13_, and its activation is linked to an outflow of calcium from intracellular stores via phospholipase C [181]. Several cannabinoid compounds have been found to bind to GPR18 and GPR55 as anandamide, 2-AG; the bioactive lipid related to endogenous cannabinoids lysophosphatidylinositol (LPI); the phytocannabinoids THC and CBD; and the synthetic compounds CP 55,940, AM251 [181,182,183]. Both GPR18 and GPR55 form a receptor-receptor interaction with CB2 in microglia [184,185]. The physiological properties of this heteroreceptor are not still fully elucidated; however, some studies showed a negative cross-talk between GPR55 and CB2 [185,186]. GPR55 modulates neuroinflammation, and its activation has been reported to increase the release of interleukins (ILs) [187]. On the other hand, GPR55 antagonists effectively block microglial activation, similarly in GPR55^−/−^ knockout mice have observed a reduction in the release of the proinflammatory cytokines [188,189]. GPR55^−/−^ mice show a normal life span and no alteration in endocannabinoids and related lipids levels; however, a deficit in motor coordination was reported, supporting a role for GRP55 in motor function [190]. GPR55 is highly expressed in the hippocampus, which suggests a role in learning and memory processes. The pharmacological inhibition of GPR55 has been associated with an improvement in spatial learning and memory in rats [191]. In a recent study, GPR55 was found highly expressed in the hippocampus dentate gyrus, CA1, and CA3 of the 5xFAD AD mouse model [192]. These findings, taken together, support GPRs being potentially involved in AD pathology and can be considered promising novel pharmacological targets for AD treatment. In particular, despite just a few studies are available to date, and GPR antagonism might be associated with side effects mostly in motor function as for GPR55 deletion, the GPR modulation of inflammatory response could be a new therapeutic opportunity to counteract AD neuroinflammation.

## 4. Limits of Cannabinoids in Alzheimer’s Disease Therapy

To date, cannabis and cannabis-derived compounds have not been approved by the US Food and Drug Administration (FDA) to treat or manage Alzheimer’s, and only a few clinical trials to evaluate the use of THC (dronabinol and nabilone) or CBD have been completed or are ongoing. For example, nabilone, a synthetic cannabinoid currently approved for the treatment of chemotherapy-related nausea and vomiting, was found effective in reducing symptoms of agitation and aggression among AD patients [79]. However, to ensure patient safety, it becomes critically important to closely monitor side effects such as sedation and possibly cognitive decline. 

Considering cannabinoids as a therapeutic option, identifying an effective dosage and treatment time is challenging. It is already well-known that molecular changes related to AD began several years before symptoms manifest. As a result, neuroprotective and immunomodulatory potential effects of cannabinoids should be administered before AD is exacerbated and prolonged in time. However, studies on the long-term effects of cannabinoids are not yet available. While studies on the long-term cognitive effects of heavy cannabis use suggest, cannabis negatively influences cognitive functions, such as episodic memory, attentional control, and motor inhibition [193,194]. For this reason, further studies to explore the short- and long-term effects of cannabinoids are urgently needed.

Unfortunately, studies investigating cannabinoid drug-drug interactions are still limited. Several investigations would be fundamental to underpinning this critical point, considering patients with dementia take multiple medications, and cannabinoids could be included as an additional therapeutic strategy to tackling the symptoms of dementia.

## 5. Final Remarks

Marijuana and cannabinoids have been associated with a wide range of beneficial pharmacological effects from one side and with harmful and adverse effects from the other. The mechanisms behind this opposed phenomenon are not fully understood. However, emerging data suggest that dosage and user age are crucial factors involved in multifaceted cannabinoids effects. Marijuana’s adverse effects are mainly related to interfering with cognitive and executive functions. Many western countries are legalizing the use of marijuana without giving any education and information to users about the risks associated with its abuse. An open market makes cannabis easily accessible, increases consumption, and consequently leads to adverse health repercussions in individuals in vulnerable categories such as adolescents and young adults. 

On the contrary, increasing scientific evidence supports that the ECS is associated with neurodegenerative diseases, and modifying its tone could be a promising therapeutic tool for treating AD. In some cases, the same substances implicated in impairing learning and memory functions could be beneficial in counteracting neurodegenerative processes at low doses. Cannabinoids can reduce oxidative stress and excitotoxicity, amyloid plaques, and neurofibrillary tangles formation. AD neuroinflammatory processes can be suppressed by the immunomodulatory effect of the CB2 receptor controlling microglial activity. Another significant effect is on the availability of acetylcholine and prevention of acetylcholinesterase-induced Aβ aggregation. Most importantly, accumulated evidence indicates that cannabinoids induce neurogenesis in the hippocampus in adults. Likewise, the inhibition of endocannabinoid degradation can be a promising pharmacological strategy to counteract the aging process and have a beneficial impact on AD progression. The modulation of production and degradation of endocannabinoids can be other than efficacious, with low side effects, compared to synthetic CBs receptors agonist/antagonist compounds. Clinical studies reported several beneficial effects in AD-related symptoms after cannabinoid administration. After dronabinol consumption, patients in the late stages of dementia showed a reduction in nocturnal motor activity and agitation. Notably, CBD has shown relevant high safety and anti-AD properties by mediating mechanisms related to the non-canonical cannabinoid receptor, making it one of the most prominent candidates between the phytocannabinoid compounds to be further tested in clinical trials. Recently, spray cannabinoid-based drugs such as Sativex (containing a 1:1 ratio of THC:CBD) and Epidiolex (containing only CBD) have been approved for chronic pain conditions in the USA, Canada, and several European countries, which makes it easy to control the cannabinoid dose delivery if compared to smoke inhalation [195]. In addition, this mouth spray and oral delivery approach could be especially beneficial for individuals with AD.

## 6. Conclusions

The last in vitro and in vivo studies strongly supported the further investigation into the use of cannabinoids as a therapeutic approach to AD. Currently, only a few clinical trials have been performed. Therefore, a deeper investigation is necessary to evaluate the safety, pharmacokinetic, pharmacodynamic, and most importantly, the efficacy of cannabinoid-based drugs for treating AD.

## Figures and Tables

**Figure 1 biology-10-00542-f001:**
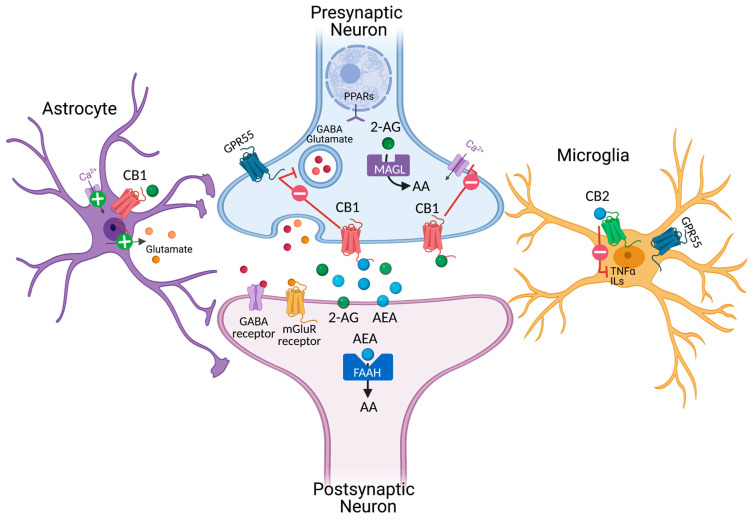
Schematic representation of the endocannabinoidergic system in the brain. Putative localization of endocannabinoid receptors in the nervous and glia system. Enzymes involved in endocannabinoid biosynthesis and degradation are reported in both pre-and postsynaptic neurons. 2-AG (green) and AEA (blue) are synthesized from phospholipids on demand. Activation of presynaptic CB1 receptors negatively modulates cell calcium influx and the release of GABA and glutamate neurotransmitters in GABAergic and glutamatergic neurons, respectively. Instead, the stimulation of CB1 in astroglia positively modulates calcium influx and glutamate release. Activation of CB2 in microglia negatively affects the release of TNFα and ILs. AA: arachidonic acid; 2-AG: 2-acylglycerol; AEA: anandamide; PPARs: peroxisome proliferator-activated receptors; FAAH: Fatty acid amide hydrolase; MAGL: monoacylglycerol lipase; mGluR metabotropic glutamate receptors; ILs: interleukins; TNFα: tumor necrosis factor-α.

**Figure 2 biology-10-00542-f002:**
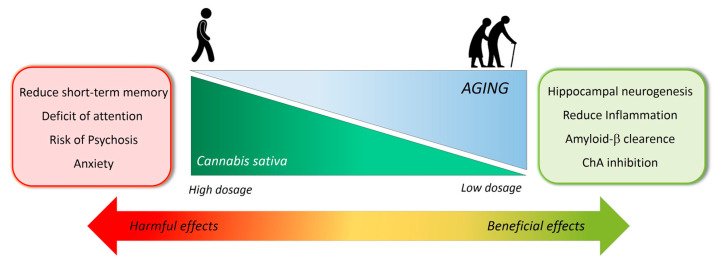
Schematic representation of the biphasic effects of THC.

## Data Availability

Not applicable.

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
