# Peer review of "Potential and Limits of Cannabinoids in Alzheimer’s Disease Therapy"

_biology, 2021, doi:10.3390/biology10060542_

Round 1

Reviewer 1 Report

This review summarizes current information regarding use of cannabinoids and modulators of cannabinoid receptors and enzymes in memory processes and Alzheimer’s Disease. The manuscript is well designed and contains updated research data. Some minor points to be addressed:

  1. Figure 1 footnote should be extended and more explicative.
  2. Although mentioned, novel orphan cannabinoid receptors such as GPR55 are somehow neglected in this review.  Some recent studies point out towards a role of these receptors in neuroinflammation and memory and a potential target in in AD. The text would be enriched with a section about these. See some literature here:
  • Saliba, S.W., Jauch, H., Gargouri, B. et al.Anti-neuroinflammatory effects of GPR55 antagonists in LPS-activated primary microglial cells. J Neuroinflammation 15, 322 (2018).
  • Medina-Vera, D.; Rosell-Valle, C.; López-Gambero, A.J.; Navarro, J.A.; Zambrana-Infantes, E.N.; Rivera, P.; Santín, L.J.; Suarez, J.; Rodríguez de Fonseca, F. Imbalance of Endocannabinoid/Lysophosphatidylinositol Receptors Marks the Severity of Alzheimer’s Disease in a Preclinical Model: A Therapeutic Opportunity. Biology20209, 377.
  • Marichal-Cancino, B. A., Fajardo-Valdez, A., Ruiz-Contreras, A. E., Méndez-Díaz, M., and Prospéro-García, O. (2018). Possible role of hippocampal GPR55 in spatial learning and memory in rats. Acta Neurobiol. Exp.
  • Wu, C.-S., Chen, H., Sun, H., Zhu, J., Jew, C. P., Wager-Miller, J., et al. (2013). GPR55, a G-protein coupled receptor for lysophosphatidylinositol, plays a role in motor coordination. PLoS One8:e60314.
  1. The mechanism through which CB1 is related to memory loss is likely related to mTORC1 activation and protein synthesis. This may be included in the text. Some literature regarding this issue here:
  • Sharma A, Hoeffer CA, Takayasu Y, et al. Dysregulation of mTOR signaling in fragile X syndrome. J Neurosci. 2010;30(2):694-702.
  • Ehninger D, Han S, Shilyansky C, et al. Reversal of learning deficits in a Tsc2+/− mouse model of tuberous sclerosis. Nat Med. 2008;14 (8):843-848

Author Response

Reviewer 1:

This review summarizes current information regarding use of cannabinoids and modulators of cannabinoid receptors and enzymes in memory processes and Alzheimer’s Disease. The manuscript is well designed and contains updated research data.

We would like to thanks the reviewer for the kind words and for the appreciations of our review and for the useful suggestions. We have addressed all the raised aspects and we have modified the manuscript accordingly.

Some minor points to be addressed:

  1. Figure 1 footnote should be extended and more explicative.

REPLY: We completely agree with the reviewer suggestion, and we apologize for the missing information on figure 1. A more detailed figure legend has been included and reported as follow:

“Figure 1. Schematic representation of the endocannabinoidergic system in the brain. Putative localization of endocannabinoid receptors in the nervous and glia system. Enzymes involved in endocannabinoid biosynthesis and degradation are reported in both pre-and postsynaptic neurons. 2-AG (green) and AEA (blue) are synthesized from phospholipids on demand. Activation of presynaptic CB1 receptors negatively modulates cell calcium influx and the release of GABA and glutamate neurotransmitters in GABAaergic and glutamatergic neurons, respectively. Instead, the stimulation of CB1 in astroglia positively modulates calcium influx and glutamate release. Activation of CB2 in microglia negatively affects the release of TNFα and ILs. AA, arachidonic acid; 2-AG, 2-acylglycerol; AEA, anandamide; PPARs, peroxisome proliferator-activated receptors; FAAH, Fatty acid amide hydrolase; MAGL, monoacylglycerol lipase; mGluR metabotropic glutamate receptors; ILs, interleukins; TNFα, tumor necrosis factor-α.”

  1. Although mentioned, novel orphan cannabinoid receptors such as GPR55 are somehow neglected in this review.  Some recent studies point out towards a role of these receptors in neuroinflammation and memory and a potential target in in AD. The text would be enriched with a section about these. See some literature here:

  • Saliba, S.W., Jauch, H., Gargouri, B. et al. Anti-neuroinflammatory effects of GPR55 antagonists in LPS-activated primary microglial cells. J Neuroinflammation15, 322 (2018).
  • Medina-Vera, D.; Rosell-Valle, C.; López-Gambero, A.J.; Navarro, J.A.; Zambrana-Infantes, E.N.; Rivera, P.; Santín, L.J.; Suarez, J.; Rodríguez de Fonseca, F. Imbalance of endocannabinoid/ Lysophosphatidylinositol Receptors Marks the Severity of Alzheimer’s Disease in a Preclinical Model: A Therapeutic Opportunity. Biology20209, 377.
  • Marichal-Cancino, B. A., Fajardo-Valdez, A., Ruiz-Contreras, A. E., Méndez-Díaz, M., and Prospéro-García, O. (2018). Possible role of hippocampal GPR55 in spatial learning and memory in rats. Acta Neurobiol. Exp.
  • Wu, C.-S., Chen, H., Sun, H., Zhu, J., Jew, C. P., Wager-Miller, J., et al. (2013). GPR55, a G-protein coupled receptor for lysophosphatidylinositol, plays a role in motor coordination. PLoS One8:e60314.

REPLY: We thank reviewer for his/her very pertinent comments. Now a new chapter dedicated to the orphan G-protein-coupled receptors (GPRs) have been included in the revised version of the manuscript. Also suggested references have been included when appropriate.

  1. The mechanism through which CB1 is related to memory loss is likely related to mTORC1 activation and protein synthesis. This may be included in the text. Some literature regarding this issue here:

  • Sharma A, Hoeffer CA, Takayasu Y, et al. Dysregulation of mTOR signaling in fragile X syndrome. J Neurosci. 2010;30(2):694-702.
  • Ehninger D, Han S, Shilyansky C, et al. Reversal of learning deficits in a Tsc2+/− mouse model of tuberous sclerosis. Nat Med. 2008;14 (8):843-848

REPLY: We thank reviewer for his/her comments. The activation of mTORC1 was found able to enhances Aβ generation and deposition by modulating amyloid precursor protein (APP) metabolism and upregulating β-, γ-secretases and aberrant hyperphosphorylated tau. Thus inhibiting the activation of mTOR is an important therapeutic target for AD. In light of these, in literature the specific modulation and the direct link between CB1 (and also CB2) and mTORC1 in AD still need to be further explored. Thus, now the manuscript included the suggested references and has been modified as follow:

Another potential of endocannabinoids as a therapeutic option for AD, is their ability to modulate the mammalian target of the rapamycin (mTOR) signaling pathway [162,163]. The activation of mTOR is a trigger for Aβ generation, thus its inhibition is an important therapeutic target for AD [164]. Of note, 2-AG treatment was able to prevent the activation of mTOR signaling pathway in the hippocampus in mice through a CB2-dependent mechanism [165]. Again, CB1 and mTOR are intimately linked and involved in regulating of excitatory glutamatergic inputs and energy balance at the brain level [166]. Overall, despite this intriguing link between endocannabinoids and mTOR need to be further explored, these data further confirmed the endocannabinoid system as an attractive therapeutic strategy to be further deepened in AD.”

Reviewer 2 Report

The present review entitled “Potential and Limits of Cannabinoids in Alzheimer’s disease therapy” by Abate et. al. summarizes the current knowledge on the potential and risks of cannabinoid usage for the treatment of Alzheimer’s disease (AD).

The review is very well written, discusses all the major aspects concerning natural occurring contents of cannabis sativa, synthetic cannabinoids as well as the modulation of endocannabinoids.

I have no major concerns regarding this article and strongly recommend its publication after minor revision.

Specific comments:

I have noticed some minor mistakes:

Line 47: preselinin-1 instead of presenilin-1  

132: … one of the highest expressed? Or most abundant?

Line: 265 symbols missing

Line: 267 symbol missing

Line: 278 symbol missing

A suggestion of minor changes:

Figure 1 should be shortly explained in a figure legend.

Mistakes in sentence structure:

Line 141-143

Line: 189-191

Line: 289-293

Line: 481-483

Author Response

REVIEWER 2

The present review entitled “Potential and Limits of Cannabinoids in Alzheimer’s disease therapy” by Abate et. al. summarizes the current knowledge on the potential and risks of cannabinoid usage for the treatment of Alzheimer’s disease (AD).

The review is very well written, discusses all the major aspects concerning natural occurring contents of cannabis sativa, synthetic cannabinoids as well as the modulation of endocannabinoids.

I have no major concerns regarding this article and strongly recommend its publication after minor revision.

We really appreciate the reviewer for his/her valuable comments and for his/her interesting in our manuscript. We have now addressed all the raised aspects and modified the manuscript accordingly. Now the quality of the manuscript has been improved thanks for reviewer’s comments.

Specific comments:

I have noticed some minor mistakes:

Line 47: preselinin-1 instead of presenilin-1

132: … one of the highest expressed? Or most abundant?

Line: 265 symbols missing

Line: 267 symbol missing

Line: 278 symbol missing

REPLY: We apologize for the editorial errors found during the review. We have modified the manuscript accordingly and now all these suggestions have been included.

A suggestion of minor changes:

Figure 1 should be shortly explained in a figure legend.

REPLY: A more detailed figure legend for Figure 1 has been included and reported as follow:

“Figure 1. Schematic representation of the endocannabinoidergic system in the brain. Putative localization of endocannabinoid receptors in the nervous and glia system. Enzymes involved in endocannabinoid biosynthesis and degradation are reported in both pre-and postsynaptic neurons. 2-AG (green) and AEA (blue) are synthesized from phospholipids on demand. Activation of presynaptic CB1 receptors negatively modulates cell calcium influx and the release of GABA and glutamate neurotransmitters in GABAaergic and glutamatergic neurons, respectively. Instead, the stimulation of CB1 in astroglia positively modulates calcium influx and glutamate release. Activation of CB2 in microglia negatively affects the release of TNFα and ILs. AA, arachidonic acid; 2-AG, 2-acylglycerol; AEA, anandamide; PPARs, peroxisome proliferator-activated receptors; FAAH, Fatty acid amide hydrolase; MAGL, monoacylglycerol lipase; mGluR metabotropic glutamate receptors; ILs, interleukins; TNFα, tumor necrosis factor-α.”

Further, Figure 1 now has also been better included in manuscript chapter as follow:

“As described in Figure 1 that reported a schematic representation of the endocannabinoidergic system at neuronal level, endocannabinoids are released in the synaptic cleft from the postsynaptic neurons. They interact with the cannabinoid receptors located on the presynaptic neurons, negatively modulating the GABA and glutamate release. [41].” 

Mistakes in sentence structure:

Reply: We apologize for this, we have now changed the sentences:

Line 141-143

Now reported in line 152-154: “Several findings showed that acute activation of CB1, especially at a young age, negatively affects dose-dependently short-term memory performance”.

Line: 189-191

Now reported in line 197-199: “THC reduced the fluorescence intensity in the Thioflavin test in a dose-dependent manner by direct interaction with the Aβ peptide [58], affecting Aβ fibril formation and aggregation  [59].

Line: 289-293

Now reported in line 297-301: “The negative effects of synthetic cannabinoids (WIN 55,212-2 and CP 55,940) on learning and memory appear to be directly linked to the inhibition of acetylcholine release in the hippocampal region [100,101] and the inhibition of glutamatergic synaptic transmission in the prefrontal cortex [102,103]. 

Line: 481-483

Now reported in line 541-544: To date, cannabis and cannabis-derived compounds are not approved by the U.S. Food and Drug Administration (FDA) to treat or manage Alzheimer's, and only a few clinical trials to evaluate the use of THC (Nabilone) or CBD have been completed or are ongoing.”